# Estimating Position from Millisecond Samples of GPS Signals (the “FastFix” Algorithm)

**DOI:** 10.3390/s20226480

**Published:** 2020-11-13

**Authors:** Timothy C. A. Molteno

**Affiliations:** Department of Physics, University of Otago, Dunedin 9016, New Zealand; tim@physics.otago.ac.nz

**Keywords:** GPS, GNSS, navigation, codephase, wildlife tracking

## Abstract

A new approach to GPS positioning is described in which the post-processing of ultra-short sequences of captured GPS signal data can produce an estimate of receiver location. The algorithm, called ‘FastFix’, needs only 2–4 ms of stored L1-band data sampled at ∼16 MHz. The algorithm uses a least-squares optimization to estimate receiver position and GPS time from measurements of the relative codephase, and Doppler-shift of GNSS satellite signals. A practical application of this algorithm is demonstrated in a small, lightweight, low-power tracking tag that periodically wakes-up, records and stores 4 ms of GPS L1-band signal and returns to a low-power state—reducing power requirements by a factor of ∼10,000 compared to typical GPS devices. Stationary device testing shows a median error of 27.7 m with a small patch antenna. Results from deployment of this tag on adult Royal Albatross show excellent performance, demonstrating lightweight, solar-powered, long-term tracking of these remarkable birds. This work was performed on the GPS system; however, the algorithm is applicable to other GNSS systems.

## 1. Introduction

Tracking devices based on GNSS (Global Navigation Satellite System) are now commonplace, their many uses include wildlife tracking [1] and vehicle tracking [2]. Wildlife applications, in particular, are constrained by size and weight, and, until recently, GNSS devices have been limited to tracking animals such as moose [3] and marine species [4] capable of carrying a large receiver, limiting the range of species that can be tagged [5]. Improvements in semiconductor fabrication have steadily reduced power consumption [6], and improvement in battery technology [7] have reduced the weight of such tracking devices to the point that GNSS tracking devices can be deployed to track small mammals and medium-sized birds [8,9]. With each technical development allowing smaller and lower-power tracking devices, the range of species that can be tracked, and the duration of the deployment is increased.

The most widely used GNSS is the GPS (global Positioning System) network [10]. GNSS tracking devices operate by using codephase and navigation message data transmitted from the satellites to determine the time of transmission of a signal. This can be used to determine the distance (ranges) from the current position to each satellite as well as the precise local time (see for example Kaplan [10]).

Navigation messages transmitted from the receiver contain sufficient information to determine orbital parameters (ephemerides) of the satellites and the time of propagation of the signals from the satellite, and hence the ranges. Using the measured ranges from four or more satellites, the position of the receiver is determined—commonly using a least-squares procedure [11]. The time to first fix typically exceeds 30 s, as this is the time required for all of the satellite ephemerides to be transmitted. As a result, traditional GPS receivers are required to operate from cold-start for more than 30 s—typically consuming tens of milliamps—in order to measure their position. GNSS position accuracy varies, and in the case of GPS, the distribution of positioning errors is not Gaussian [12] and depends sensitively on the antenna characteristics [10]; typically, the first fix from a single-frequency commercial GPS has a horizontal positioning error of ∼10 m [13] in open skies.

### Alternative Algorithms

Advances in algorithms provide another technique to reduce power consumption, and several alternative approaches to the traditional positioning algorithm have been developed. Of these, the most widely used is Assisted GPS (A-GPS). This employs satellite ephemeris data, and estimates of position, time to eliminate the need to receive this data via the GNSS signal [14]. This has limitations in tracking applications as a separate telemetry channel is needed for the augmentation data. Time-to-first-fix from a cold start situation, signal sensitivity, and position-accuracy are all improved with respect to a stand-alone GPS receiver [15].

Another algorithm family estimate position using only millisecond samples of data, with the satellite ephemeris provided externally. This technique is referred to as time-free, or snapshot positioning and was first proposed by Peterson et al. in 1995 [16]. This was refined in 2009 [17] to show that positioning was possible using only codephase measurements. Van Dierendonck et al. [18] was proposed in 2015 that uses an initial approximation of receiver time and position, to reconstruct the transmit time relative to the received time such that pseudorange can be calculated, and position estimated. In 2012, Othieno and Gleason [19] introduced a positioning method that use a combination of Doppler and code-phase measurements to provide a position estimate unknown initial position. Jing et al. [20] in 2017 introduced an integer ambiguity searching method based on the relationship between the maximum pseudorange difference and the inter-satellite distance. The accuracy possible with these snapshot algorithms is typically ∼20 m [21].

This algorithm described in this paper uses Doppler information to provide an a priori position estimate, in a manner similar to Othieno [19], and combines a relative-codephase measure, in a manner analogous to the work of Jing et al. [20] who instead use an inter-satellite distance. In practice, the algorithm is able to determine position anywhere on earth from as little as 2 ms of GPS L1-band signal using measurements of Doppler shift and *relative codephase* recovered from the data. Section 2 describes the well-known process of *acquisition*, and shows that small errors in the knowledge of the local time at the receiver will affect the measurement of codephase; however, this effect is the same for each satellite signal, and introduces relative codephase which is not affected by these errors. Section 3 describes the FastFix algorithm for positioning, and Section 4 presents results for stationary device performance including typical accuracy. Finally, Section 5 presents a real-world application of this algorithm in tracking albatross.

## 2. Satellite Signal Acquisition

The GPS L1-band civilian signal is transmitted in a 2.5 MHz bandwidth radio channel centered at 1575.42 MHz. The satellite signals are modulated using Code Division Multiple Access (CDMA) [10]. A GPS receiver digitizes this channel, and signals from all visible satellites are contained within the received data. The individual satellite signals are recovered by correlating the received data against a pseudorandom sequence unique to each satellite called a C/A (Coarse Acquisition) code. In the case of GPS, the C/A code repeats every millisecond and contains 1023 binary numbers in a pseudorandom sequence. The motion of each satellite relative to the observer causes the signal from that satellite to be Doppler-shifted, and this Doppler-shift must be compensated for in the receiver. The process of *signal acquisition* recovers the signal, and its Doppler-shift from each visible satellite.

There are several well-established methods for signal acquisition; see, for example, Kaplan [10] or Borre et al. [22]. In this work, parallel codephase recovery [22] using Fourier transforms was used. For the *i*th satellite, the acquisition process results in three measurements:$X_{max}/\sigma_{X}$. The maximum correlation divided by the standard-deviation of the correlation—a measure of the ratio of correlation amplitude to the noise. This is similar to the carrier-to-noise density ratio, or the correlation amplitude relative to the noise level in the data. A satellite is considered *visible* if its $X_{max}/\sigma_{X}$ exceeds a threshold (for the data shown in Figure 1 and Table 1, a threshold of 7 is used).$\delta_{\phi }^{i}$, the *code-phase*—a number between zero and 1 that indicates the phase of the C/A code. The starting time of the C/A code is given by $t_{0}+\delta_{\phi }^{i}\tau$, where $t_{0}$ is the time of the start of the data, and τ is the C/A code sequence repeat time (one millisecond in the case of GPS).$\Delta f_{m}^{i}$, the Doppler-shift of the signal from the satellite.

Table 1 shows a typical set of acquisition results from the data, which are shown in Figure 1a. Some of the correlation plots from this data are shown in Figure 1b–d.

### 2.1. System and Local Time Frames

The acquisition process uses data that starts at time $t_{0}$, measured by the receiver clock that may be inaccurate. Times measured by the receiver are identified by lower case *t*. The GNSS system clock times are identified by upper-case time values and are typically accurate to better than 1 μs [10].

All GNSS satellites synchronously send their signals at intervals of τ. The start of each code epoch is synchronous across all satellites in the constellation. The system time that the *i*th satellite signal is sent, $T_{s}^{i}$, is given by $T_{s}^{i}=T_{s}+M\tau$, where $T_{s}$ is an arbitrary start time for the GPS constellation of satellites, and *M* is the integer code epoch. We choose, quite arbitrarily, that $T_{s}=0$. Therefore, the time of transmission for the start of the code $T_{s}^{i}$ becomes
(1)$$T_{s}^{i}=M^{i}\tau ,$$
where the $M^{i}$ are integers that are now different for each satellite, and depend on the relative distance between receiver and each satellite.

We assume that there is a fixed offset, $t_{u}$, between the local receiver clock and the system time, i.e.,
(2)$$T=t+t_{u}.$$

A more sophisticated model of the receiver clock would have both an offset and a different rate, but as the data are sampled for only a few milliseconds, uncertainty in the local clock rate will not significantly affect this algorithm.

### 2.2. Signal Propagation Time

Let $T_{r}^{i}$ be the system time that the start of the *i*th C/A code reaches the receiver. This is given by
$$T_{r}^{i}=t^{i}+t_{u},$$
where $t^{i}\equiv t_{0}+\delta_{\phi }^{i}\tau$ is the measured time for the start of the sequence at the receiver, and $t_{u}$ is the receiver clock offset. The system time at the receiver for the start of the C/A code from satellite *i* is given in terms of $t_{0}$ and the code-phase as
(3)$$T_{r}^{i}=t_{0}+\delta_{\phi }^{i}\tau +t_{u}.$$

At system time $T_{r}^{i}$, the signal must have originated from the satellite at an earlier time $T_{s}^{i}=T_{r}^{i}-\frac{\Delta_{r}^{i}}{c}$ due to the propagation velocity *c* of the signal over the distance, $\Delta_{r}^{i}$, between receiver and satellite *i*. We know that $T_{s}^{i}$ is a time when an epoch starts, so it is an integer multiple of τ, i.e., $M^{i}\tau$, so, in this case, we write
(4)$$\begin{array}{ccc} M^{i}\tau & = & T_{r}^{i}-\frac{\Delta_{r}^{i}}{c},\\ & = & t_{0}+\delta_{\phi }^{i}\tau +t_{u}-\frac{\Delta_{r}^{i}}{c},\\ \delta_{\phi }^{i}\tau & = & M^{i}\tau +\frac{\Delta_{r}^{i}}{c}-\left(t_{0}+t_{u}\right). \end{array}$$

The codephase $\delta_{\phi }^{i}$ changes rapidly with time due to satellite motion, i.e, $\frac{d\delta_{\phi }^{i}}{dt}=\frac{\Delta_{r}^{i}}{c\tau }∼67$ cycles per second. In addition, uncertainty in the system time offset, of more than a fraction of a millisecond will make prediction of position from the codephase unfeasible. For example, an uncertainty in $t_{u}$ of 100 μs would lead to an uncertainty in $\delta_{\phi }^{i}$ of 0.1, which would translate to a pseudorange error of ∼30 km.

### 2.3. Relative Fractional Codephase

The relative fractional codephase, $\delta_{\phi }^{ij}$ is the difference (modulo τ) between the fractional codephases $\delta_{\phi }^{i}$ and $\delta_{\phi }^{j}$ from satellite *i* and satellite *j*. From Equation (Equation 4),
$$\begin{array}{ccc} \delta_{\phi }^{i}\tau -\delta_{\phi }^{j}\tau & = & \left(M^{i}-M^{j}\right)\tau +\left(\frac{\Delta_{r}^{i}}{c}-\frac{\Delta_{r}^{j}}{c}\right)\\ \delta_{\phi }^{i}-\delta_{\phi }^{j} & = & \left[\left(M^{i}-M^{j}\right)+\left(\frac{\Delta_{r}^{i}-\Delta_{r}^{j}}{c\tau }\right)\right]\\ \delta_{\phi }^{i}-\delta_{\phi }^{j} & = & N^{ij}+\delta_{\phi }^{ij}, \end{array}$$
where $N^{ij}=M^{i}-M^{j}$ is the integer part of the expression, and the relative codephase $\delta_{\phi }^{ij}$ is the fractional part of the expression. For example,
(5)$$\begin{array}{ccc} \delta_{\phi }^{ij}\left(t,r_{0}\right) & = & \mathrm{frac}\left(\frac{\Delta_{r}^{i}-\Delta_{r}^{j}}{c\tau }\right). \end{array}$$

This relative fractional-codephase does not depend sensitively on accurate knowledge of the local time offset $t_{u}$; however, range errors due to incorrect time estimates will have an effect. Errors of 50 km or more will still lead to relatively small changes in $\delta_{\phi }^{ij}$. At typical satellite velocities of ∼3 km s${}^{-1}$, an uncertainty of ten seconds in $t_{u}$ would lead to a typical $\delta_{\phi }^{ij}\left(t,r_{0}\right)$ uncertainty of approximately 0.1. Contrast this with the codephase $\delta_{\phi }^{i}$, for which this level of uncertainty would be caused by a ∼100 μs uncertainty in $t_{u}$.

## 3. The FastFix Positioning Algorithm

The FastFix algorithm uses both the Doppler-shift and the codephase measurements to solve the inverse problem of estimating the position, $r_{0}$, and time-of-fix, $t_{0}$ from stored GPS $L_{1}$-band data. The forward problem assumes that the position r, local time $t_{0}$ and local clock offset $t_{u}$ are known, and satellite ephemerides are available. This is sufficient to determine from Equation (Equation 5) the relative codephase, $\delta_{\phi }^{ij}$, and the Doppler-shift of the signal from the *i*th satellite,
(6)$$\Delta f^{i}\left(t_{0},r\right)=-\frac{1}{c}\frac{d\Delta_{r}^{i}}{dt}f_{0},$$
where $f_{0}∼1575.42$ MHz, *c* is the speed of light, and $\Delta_{r}^{i}$ is the relative distance between the *i*th satellite and the receiver position r. If $\Delta_{r}^{i}$ is decreasing, then satellite is moving towards the receiver, and the Doppler-shift is positive.

This method is rather like the method of exact fractions from interferometry (see, for example, Baird [23]), the measurements comprise only the fractional part of the codephase for the signal from each satellite and its Doppler shift. At any given moment, there are only a small number of positions near the earth’s surface that will have any specific set of relative-codephases.

### 3.1. Least-Squares Estimators

The algorithm uses two least-squares estimators: one based on Doppler measurements, the other based on the relative fractional codephase. The least-squares estimator for Doppler shift, $L_{\nu }$, is
(7)$$L_{\nu }\left(t,r_{0}\right)=\sum_{i}{\left(\Delta f_{m}^{i}-\Delta f^{i}\left(t,r_{0}\right)\right)}^{2},$$
where $\Delta f_{m}^{i}$ is the measured Doppler-shift for the *i*th satellite, and $\Delta f^{i}\left(t,{\dot{r}}_{0}\right)$ is the predicted Doppler-shift for the *i*th satellite from Equation (Equation 6). The least-squares measure for relative fractional codephase, $L_{\delta \phi }$, is
(8)$$L_{\delta \phi }\left(t,r_{0}\right)=\sum_{i,j}{\left(\delta_{\phi m}^{ij}-\delta_{\phi }^{ij}\left(t,r_{0}\right)\right)}^{2},$$
where $\delta_{\phi m}^{ij}$ is the measured relative fractional codephase between the *i*th and *j*th satellite, and $\delta_{\phi }^{ij}\left(t,r_{0}\right)$ is the predicted relative fractional codephase given by Equation (Equation 5).

### 3.2. Algorithm Outline

The algorithm begins with an estimate of the local time $t_{0}$, and, by acquiring the codephase, Doppler-shift and $X_{max}/\sigma_{X}$ for each visible satellite from the previously stored GPS $L_{1}$-data. These measurements are then used for three successive least-squares estimations of position. Starting with a global minimization based on Doppler-shift measurement, the successive stages each further refine the position from the preceding stage. A flow-chart for the algorithm is shown in Figure 2. Each of the minimization stages requires knowledge to the satellite orbital elements, and it is assumed that these are downloaded separately from a suitable catalog.

The sensitivity of the $\Delta f^{i}$ to the solution is quite low, so, from an initial measurement of $t_{0}$, an initial first pass is made searching for a position that minimizes $L_{\nu }$ from Equation (Equation 7). This results in a rough position estimate $r_{1}$.

This Doppler-estimate is refined by searching in the region near $r_{1}$ for a position $r_{2}$ that optimizes the relative fractional codephase estimator, $L_{\delta \phi }$ at time $t_{0}$.

The final step is a full four-dimensional space-time optimisation using a Nelder–Meade [24] optimisation algorithm. This yields an estimate for the receiver position, $r_{3}$, as well as the local time $t_{1}$.

## 4. Results

The FastFix algorithm was initially tested using stationary receivers in a known locations (Dunedin, New Zealand). In this test, 118 four-millisecond samples of GPS L1-band data were digitized and stored. From these, the FastFix algorithm (described in the previous section) was applied.

### 4.1. Doppler Search

Figure 3 shows contours of the Doppler least-squares estimator $L_{\nu }$ plotted over the globe centered on the receiver location. This shows a global and local minimum at the receiver location. This estimate is not accurate, however it is sufficient to locate the receiver within a small enough region for the next step which is the minimisation of the relative codephase estimator $L_{\delta \phi }$. Figure 4 shows a histogram of the Doppler-only location error, showing a median error of 175 km and a 95th percentile error of 537 km.

### 4.2. Space Search

Figure 5 shows the relative codephase estimator $L_{\delta \phi }$ plotted for six visible satellites over a region ten degrees in width and height centred at the actual receiver location. This figure shows that the search for the global minimum is a complex nonlinear minimisation problem. However, the Doppler search is sufficiently accurate to place the receiver within the central local minimum region and the relative codephase least-squares estimator can be easily minimised once this is known.

### 4.3. Measured Accuracy

Figure 6 shows a histogram of the fix error for a known location. This shows that the relative codephase least-squares minimisation produces accurate fixes most of the time. The median error is 27.7 m, and all results are based on 4 ms of sampled data from a GPS front end (Maxim MAX2769B Universal GPS Receiver).

There are outliers that are caused by convergence on to a local, but not global minimum. Work is continuing on improving the performance of this; however, the results are already good enough to produce a device based on this technique. Section 4.4 outlines some techniques that may improve the robustness and accuracy of the results. This positioning error is similar to other snapshot positioning algorithms [21].

### 4.4. Discussion and Potential Improvements

One avenue for improving robustness is establishing better criteria for rejecting poor position estimates. Cook’s distance is a commonly used estimate of the influence of a data point when doing least-squares regression. Cook’s distance measures the effect of deleting a given observation. Another technique is to use Huber’s M-estimation [25] to handle outliers due to bad measurements.

#### Time-Separated Measurements

Another possible improvement would be to combine two sets of relative codephase and Doppler measurements taken from two samples of L1-band data a short time, Γ, apart. Assuming Γ is large enough for the satellites to have moved by ∼1 km, yet small enough for the receiver to have moved by less than the desired spatial resolution, then the following relative codephase estimator should be more robust,
$$\begin{array}{ccc} L_{\delta \phi }\left(t,r_{0}\right) & = & \sum_{i}{\left(\delta_{\phi m}^{ij}-\delta_{\phi }^{ij}\left(t,r_{0}\right)\right)}^{2}+\\ & & \sum_{i}{\left((\delta_{\phi m}^{ij}-\delta_{\phi }^{ij}\left(t+\Gamma ,r_{0}\right)\right)}^{2}. \end{array}$$

Taking multiple samples would, however, increase the power required.

## 5. Application to Wildlife Tracking

There are many possible applications for devices that use this algorithm. As only very short snapshots of GPS data are required, a device only needs to operate for 4 ms to acquire the data necessary for a fix. This dramatically reduces the power required (from at least 30 s of operation for a traditional GPS fix), by a factor of approximately 10,000.

A disadvantage of tracking with this technique is that any tracking device has to be recovered in order for the position to be estimated by post-processing the stored data. This technique is suitable for tracking species that return to a known location so that the data can be recovered. Here, we describe how this was done on nesting albatross.

### 5.1. Tag Electronics

Tags were developed that include a standard GPS RF front-end (Maxim MAX2769B Universal GPS Receiver) connected to a high-performance low-power 32-bit STM32F103CB [26] microprocessor. The microprocessor is programmed to wake at intervals, read data from the GPS RF front-end and store this data in a file on an micro Secure Digital (microSD) Memory Card. The un-packaged electronics from one of these FastFix tags are shown in Figure 7. A packaged solar-powered version of the tag, designed to attach to seabird feathers was developed [9] for deployment on seabirds. This tracking tag uses a small Taoglas AP.12F.07.0045 active patch antenna [27].

Each fix stores 4 ms of 1-bit baseband data sampled at 16.368 MHz, which requires 64 kb of data. A 2 GB SD-card can store approximately 244,000 fixes. A solar-powered tag could operate continuously storing data for a position estimate every five minutes for more than two years, limited only by storage limits.

### 5.2. Wildlife Tracking Results

Twelve solar-powered FastFix tags were attached to nesting adult Royal Albatross [28] (shown packaged and attached in Figure 8). While nesting, the adults carry out feeding flights of up to two weeks of duration. Of the 12 tags attached, eight were recovered undamaged, and from these eight tags, 26,042 positions were calculated. The position tracks recovered from the tags are shown in Figure 9.

## 6. Conclusions

The algorithm described here allows ultra-low-power, reasonably accurate global position determination using very short sequences of GPS data. The computational effort required to measure position is a function of both the initial uncertainty in position and the local receiver clock offset. The results in stationary testing show a median error of 27.7 m which is worse than the median error of traditional GPS single-frequency receivers, but similar to existing snapshot algorithms [21]. I speculate that this is due to the lack of a code-tracking filter that can refine the pseudorange estimates during the relatively long period that a traditional GPS device spends receiving navigation information. Despite this limitation, there are several interesting avenues to explore for improving the robustness and accuracy which were discussed in Section 4.4. The performance of the algorithm in practical moderate-precision applications such as wildlife tracking is sufficient as demonstrated in Section 5, and the power consumption was extremely low, which enabling long-term solar-powered, or primary-cell tracking of wildlife.

Another avenue for further work would be to use Bayesian inference techniques to estimate a posterior distribution of receiver positions. This would also enable uncertainty quantification and, consequently, a systematic exploration of the lower limit on the amount of stored data necessary to recover position to a given accuracy. An additional advantage of a Bayesian approach would be avoiding the decision to reject or accept a satellite as ‘visible’ from its correlation strength $X_{max}/\sigma_{X}$.

## Figures and Tables

**Figure 1 sensors-20-06480-f001:**
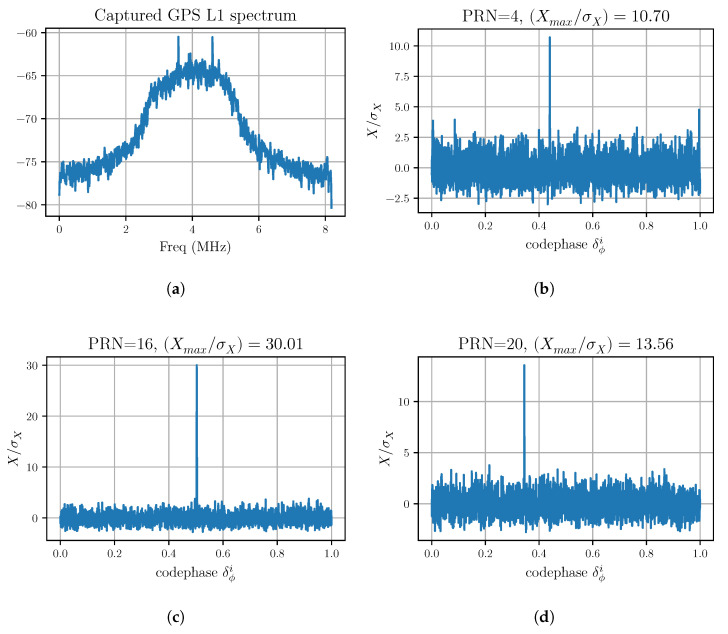
Some results showing typical correlation vs fractional-codephase for a sample of GPS L1 data, sampled at 16.368 MHz and down-sampled to 4.092 MHz. (**a**) baseband spectrum of the L1-band sampled GPS data. The center frequency is 1575.42 MHz downsampled to an intermediate frequency (IF) of 4.092 MHz. (**b**) correlation amplitude as a function of fractional codephase for the satellite with Pseudorandom Noise (PRN) code PRN = 4. The measured codephase is $\delta_{\phi }^{4}=0.4398635$; (**c**) correlation amplitude as a function of fractional codephase for satellite with PRN = 16. The measured codephase is $\delta_{\phi }^{16}=0.503052$; (**d**) correlation amplitude as a function of fractional codephase for satellite with PRN = 20. The measured codephase is $\delta_{\phi }^{20}=0.344973$.

**Figure 2 sensors-20-06480-f002:**
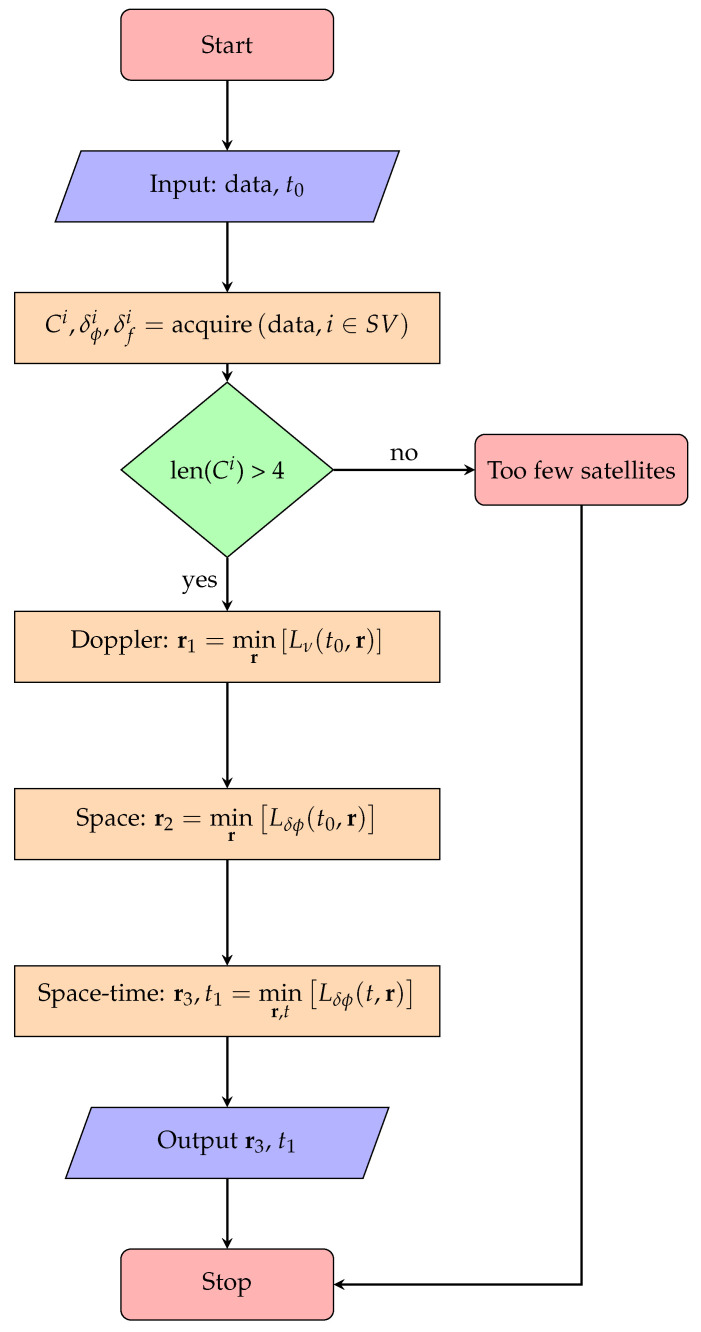
The FastFix positioning algorithm, showing the Doppler-search, space and space-time least-squares optimization processes.

**Figure 3 sensors-20-06480-f003:**
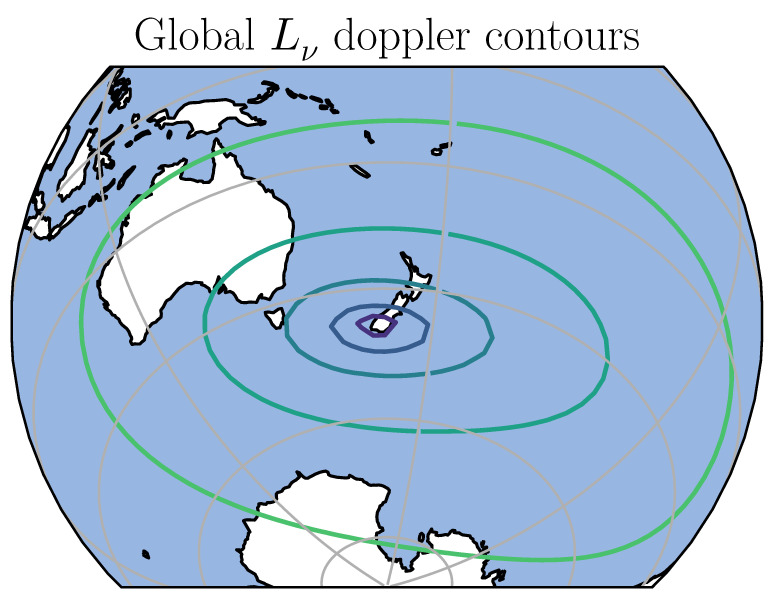
Global Doppler least-squares contours of $L_{\nu }$. This shows a global minimum close to the receiver position in New Zealand. This map is an orthographic projection of the globe, centered over New Zealand at longitude 170°, and latitude 45° south. The grey lines are 30° intervals in longitude and latitude.

**Figure 4 sensors-20-06480-f004:**
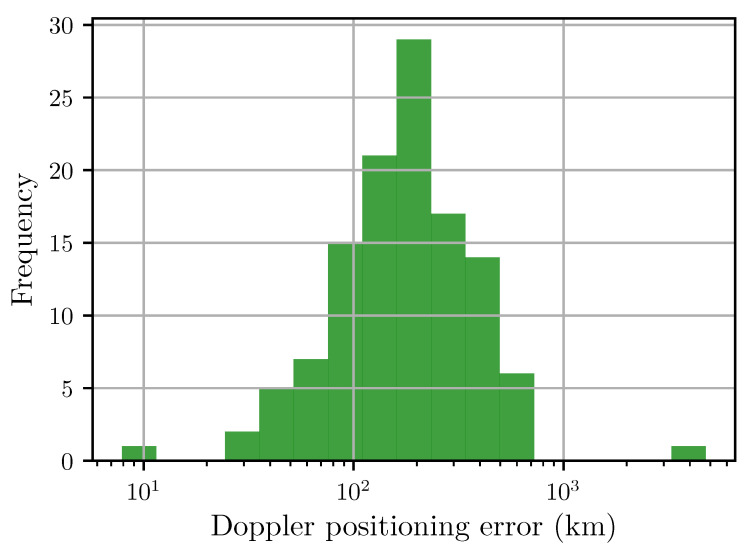
Histogram of Doppler position errors (in km) (*N* = 118). The median Doppler-only positioning error is 175 km. The 95th percentile is 537 km. The vertical axis represents the number of doppler-fixes that fall within the corresponding range of positioning error.

**Figure 5 sensors-20-06480-f005:**
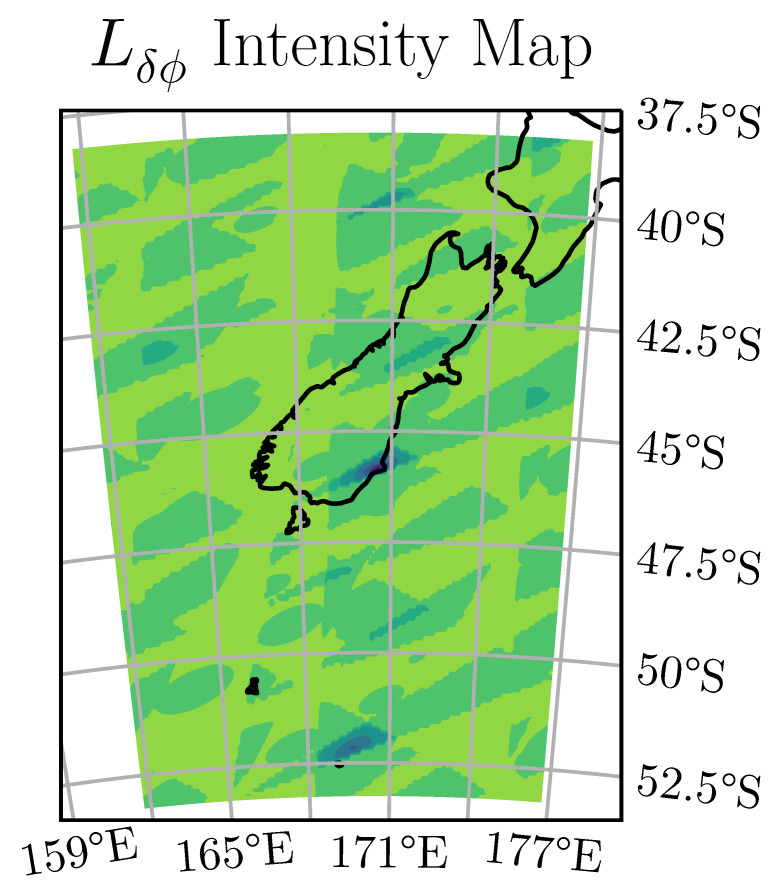
Least squares intensity map of $L_{\delta \phi }$, centered around the receiver location. Many local minima can be seen separated by approximately 300 km.

**Figure 6 sensors-20-06480-f006:**
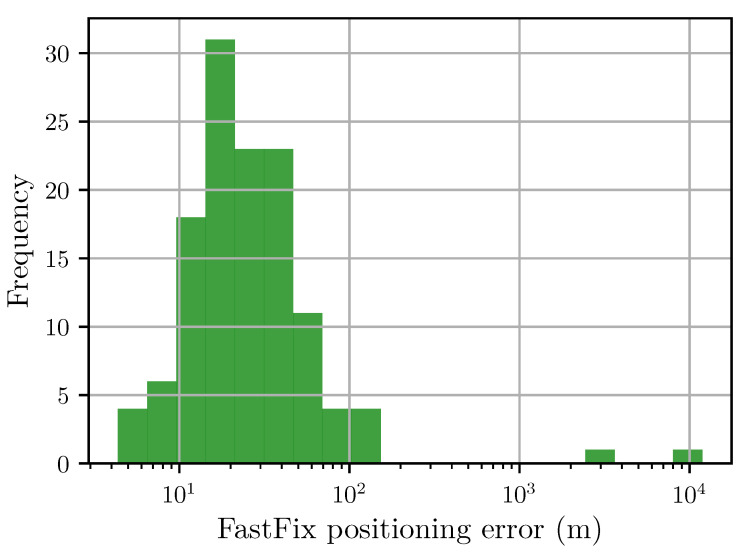
Histogram of horizontal fix errors (in meters). The median fix error is 27.7 m. The vertical axis represents the number of fixes that fall within the corresponding range of positioning error.

**Figure 7 sensors-20-06480-f007:**
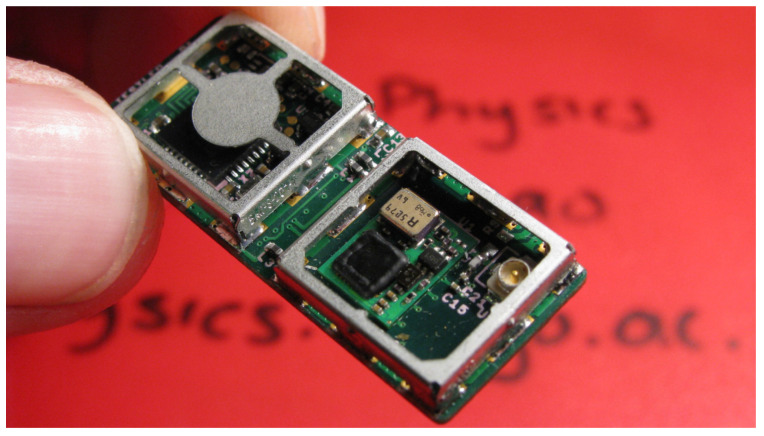
Closeup of FastFix tag electronics. The GPS RF front-end is on the right, and the microprocessor on the left. The SD card for data storage is underneath the PCB. Total tag electronics weight is 2.6 g.

**Figure 8 sensors-20-06480-f008:**
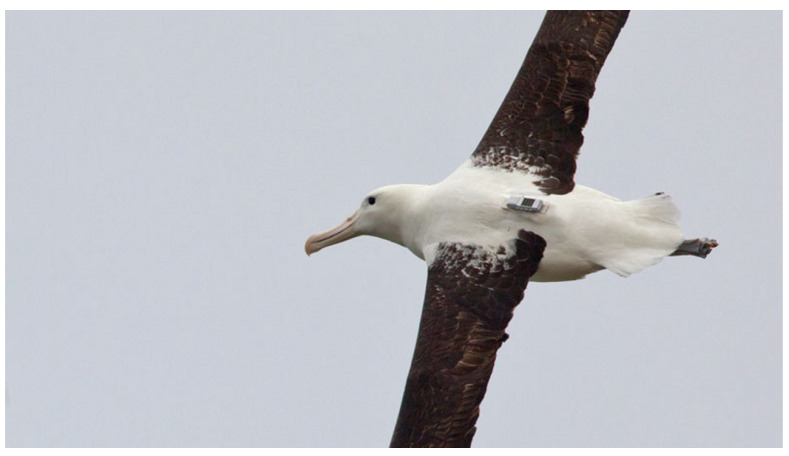
A solar-powered FastFix tag attached to the back of a Royal Albatross. This tag can store in a 2 GB SD-card approximately a quarter of a million fixes. Photo ©Keith Payne—used with permission.

**Figure 9 sensors-20-06480-f009:**
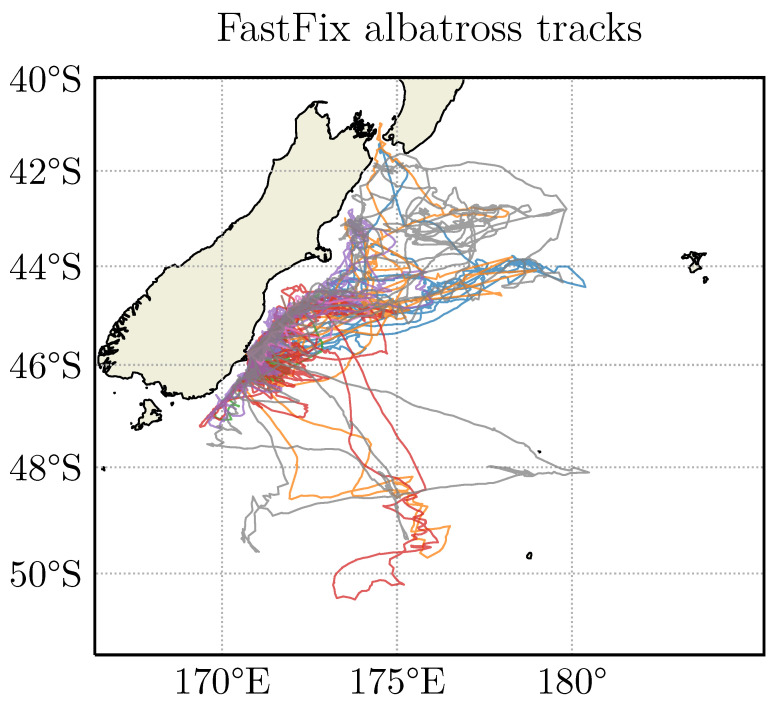
Position estimates recovered from 8 tags attached to Royal albatross over a one-month period off the coast of the South Island of New Zealand. There are 26,042 fixes. Each colour represents a different animal, showing the remarkable ability of these animals to navigate over the long distances traveled during feeding flights.

**Table 1 sensors-20-06480-t001:** Typical acquisition results for a 4 ms sample of signal (shown in Figure 1). There are six satellite signals that exceed the threshold set for signal detection in this data.

*i*	SV	$X_{\max}/\sigma_{X}$	Doppler-Shift $\Delta f^{i}$	Codephase $\delta_{\phi }^{i}$
1	16	30.01	+2185.15461	0.503051794
2	23	21.08	+1632.13669	0.576269412
3	32	16.84	−2624.85478	0.209007485
4	13	14.03	+2335.25122	0.197458456
5	20	13.56	−818.94480	0.344973267
6	4	10.70	−155.43040	0.439863498

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
