# Peer review of "Estimating Position from Millisecond Samples of GPS Signals (the “FastFix” Algorithm)"

_sensors, 2020, doi:10.3390/s20226480_

Round 1

Reviewer 1 Report

This manuscript is a good work, however, as a research paper, the authors must imporve the introduction discussing associated works published in journals during the last couple of years. This will help a reader to understand the problem and why the authors are aproposing this contribution. In addition, the reader can infer if recent and related works are improved.

The findins highlighted in the abstract: Stationary device testing shows a median error of 27 meters with a small patch
10 antenna. Results from deployment of this tag on adult Royal Albatross show excellent performance,
11 demonstrating lightweight, solar-powered, long-term tracking of these remarkable birds. This work
12 was performed on the GPS system, however the algorithm is applicable to other GNSS systems.,

they must be emphasized in the introduction and discussed with respect to similar works. A detailed review of recent literature must then be performed to have a good state of the art,

Other references must also be included, related to the hardware, for example, as done in subsection 4.1. Tag electronics
174 Tags were developed that include a standard GPS RF front-end (Maxim MAX2769B Universal
175 GPS Receiver [7]) connected to a microprocessor and an SD-card. The un-packaged electronics from
176 one of these FastFix tags are shown in Figure 7. A packaged solar-powered version of the tag, designed
177 to attach to feathers was developed [8] for deployment on seabirds. This uses a small patch antenna.

in the conclusions, it is strange dividing sub and subsubsections:

5. Conclusions
188 The algorithm described here allows ultra-low-power, reasonably accurate global position
189 determination using very short sequences of GPS data. The computational effort required to measure
190 position is a function of both the initial uncertainty in position and the local receiver clock offset. The
191 performance of the algorithm in practical moderate-precision applications such as wildlife tracking is
192 sufficient as demonstrated in Section 4. There are, however, several interesting avenues to explore for
193 improving the robustness and accuracy.
194 5.1. Potential future work
195 One avenue for improving robustness is establishing better criteria for rejecting poor position
196 estimates. Cook’s distance is a commonly used estimate of the influence of a data point when doing
197 least squares regression. Cook’s distance measures the effect of deleting a given observation. Another
198 technique is to use Huber’s M-estimation [10] to handle outliers due to bad measurements.
199 5.1.1. Time-separated measurements
200 Another possible improvement would be to combine two sets of relative codephase and Doppler
201 measurements taken from two samples of L1-band data a short time, G, apart. Assuming G is large
202 enough for the satellites to have moved by  1km, yet small enough for the receiver to have moved
Version October 14, 2020 submitted to Sensors

those subsubsections can be placed before conclusions as discussion on the experimental results and problems being solved in the near future,

Author Response

I would like to thank the reviewer for their helpful comments. I have made the changes requested, and these are itemized below (original comment in italics)

  • The findings [...] must be emphasized in the introduction and discussed with respect to similar works. A detailed review of recent literature must then be performed to have a good state of the art, A literature review section has been added to the introduction as requested that includes a review of wildlife tracking technologies, as well as a section on alternative algorithms previous work and their performance.
  • Other references must also be included, related to the hardware, for example, as done in subsection 4.1. Tag electronics. I have added references for the microprocessor in the tag, and expanded the description. Reference also added for the patch antenna.
  • in the conclusions, it is strange dividing sub and subsubsections: [...] those subsubsections can be placed before conclusions as discussion on the experimental results and problems being solved in the near future. I have removed the sub and subsubsections from the conclusions, and placed that material in a discussion section at the end of the results section.

Reviewer 2 Report

Manuscript ID: sensors-982221

Type of manuscript: Letter

Title: “Estimating position from millisecond samples of GPS signals (the “FastFix” algorithm)”

Authors: Timothy C.A. Molteno

In this work the author presents a new approach to GPS/GNSS positioning in which the post-processing of ultra-short sequences of GPS signal data can produce an estimate of receiver location with a precision of a few tens of metres. The estimation algorithm, called “FastFix”, needs only 2-4 milliseconds of stored L1-band data sampled at about 16 MHz frequency. Moreover, a portable light tag is designed and developed and applied successfully to track adult Royal Albatrosses.

The paper needs some improvements to be published on MDPI-Sensors. In particular:

As a general remark it is better to clarify all the acronyms used in the body of the text, like for example GNSS (Global Navigation Satellite System) or GPS (Global Positioning System) etc. etc..

  • Row 14 – I Think that the word “Introduction” at the beginning of this paragraph should be indicated.

  • Row 38, at the beginning of page 2, the Table 1 should be moved in the following paragraph titled “ Satellite signal acquisition” before the section 1.1.

  • At row 43 – the frequency 1.57542 GHz should be indicated as usual: 1575.42 MHz, and all frequencies should be expressed in MHz. The acronym CDMA should be clarified at least the first time is cited.

  • At row 46 – C/A code stand for Course Acquisition?

  • From row 54 to 61 are described the data represented in Table 1 – this should be indicated in the text to help the reader to understand the concept. Table 1 should be moved at the end of this section after row 63.

  • At row 58 is repeated the letter a: “a a”?

  • At row 66 of the section 1.1, I think is better to remove the word well: “clock that may well be inaccurate.”

  • At page 3 in the caption of the Figure 1(a) is better to indicate the frequency as 1575.42 MHz for compatibility with the other frequencies indicated in the following.

  • At row 95, beginning of section 2, again is preferably to indicate the frequency as 1575.42 MHz.

  • Row 110 and 113 – The proposed Algorithm should be indicated uniformly with the number 2.1 or 1. A figure with the flow chart is preferable.

  • Rows 131,132 - The Nelder-Meade optimisation algorithm needs a citation (Nelder and Meade, 1965, The Computer Journal, Volume 7, Issue 4, January 1965, Pages 308–313, https://doi.org/10.1093/comjnl/7.4.308).

  • Figure 2 needs in addition the scale and the orientation (a labelled (N-S-E-W)wind rose like figure 9 is welcome).

  • In Figure 3 is necessary to label the X and Y axis with Longitude and Latitude and it would be better to add scale and orientation.

  • From rows 152 to 154 of the section “3.3 Time search at known location” of even before, I’m not able to find the reference to the Figures 4(b) and 5 in the body of the text.

  • Rows 167 to 168 - It is true that for a traditional GPS/GNSS receiver a 30 sec fix operation is necessary to obtain the point position but the precision is in general better than few meters. Author should explain.

  • In Figure 5 and 6, if possible, the author should indicate the physical unit of the Frequency.

  • In Figure 9 a nice map is represented with the routes of Royal Albatrosses but this Figure should be presented with the scale and with the labels of the cardinal points (N-S-E-W) in the left upper side of the figure where a nice wind rose is represented. I suggest also to add a Longitude, Latitude grid with axis labels and a kilometric scale. Finally, in the right lower side of the figure is barely visible the writing: “Electronic Research Department of Physics University of Otago”.

 I therefore suggest a moderate revision.

Author Response

I would like to thank the reviewer for the helpful inputs. I will address each of their concerns below (where appropriate, I have italicised the request).

  1. I have correctly defined the GPS, GNSS, CDMA,  C/A and IF acronyms when first used.
  2. I have added a section "Introduction" as requested.
  3. The placement of the table is managed by LaTeX. I have added and [h] tag to move it as requested.
  4. I have modified 1.57542 GHz -> 1575.42 MHz on line 43 as well as the caption of Figure 1
  5. C/A has been defined as Course Acquisition (see 1) as requested.
  6. ":From row 54 to 61 are described the data represented in Table 1 – this should be indicated in the text to help the reader to understand the concept. Table 1 should be moved at the end of this section after row 63.". Table 1 has been moved, and I have reworded the reference to Table 1.
  7. The repeated 'a a' has been fixed.
  8. The word 'well' has been removed.
  9. All references to 1.57542 GHz have been changed to 1575.42 MHz as requested.
  10. The algorithm code has been replaced by a flow-chart as requested.
  11. Nelder-Meade algorithm has been referenced.
  12. In figure 2, I have indicated the scale of latitude and longitude as well as clarified that this is an orthographic projection.
  13. In Figure 3, I have added axis labels and scale.
  14. "From rows 152 to 154 of the section “3.3 Time search at known location” of even before, I’m not able to find the reference to the Figures 4(b) and 5 in the body of the text.".  I have removed the 'Time search at known location' section as I do not think it is necessary to describe the algorithm, and isn't referenced from the rest of the article.
  15. Rows 167 to 168 - It is true that for a traditional GPS/GNSS receiver a 30 sec fix operation is necessary to obtain the point position but the precision is in general better than few meters. Author should explain.
  16. In Figure 5 and 6, if possible, the author should indicate the physical unit of the Frequency. The plots are histograms, and the vertical axis is frequency of occurrence (and has no physical units). I have added a description of the y-axis in the caption.
  17. In Figure 9 a nice map is represented with the routes of Royal Albatrosses but this Figure should be presented with the scale and with the labels of the cardinal points (N-S-E-W) in the left upper side of the figure where a nice wind rose is represented. I suggest also to add a Longitude, Latitude grid with axis labels and a kilometric scale. Finally, in the right lower side of the figure is barely visible the writing: “Electronic Research Department of Physics University of Otago”. I have replaced this figure, with a new one that has scales and axis labels. Unfortunately the background bathymetry and features are not present in this new image as the software originally used is not capable of rendering coordinate grid lines.

Round 2

Reviewer 1 Report

this revised paper can be accepted now